# Impact of Gender Inequalities in the Causes of Mortality on the Competitiveness of OECD Countries

**DOI:** 10.3390/ijerph17103698

**Published:** 2020-05-24

**Authors:** Beata Gavurova, Viera Ivankova, Martin Rigelsky, Ladislav Suhanyi

**Affiliations:** 1Center for Applied Economic Research, Faculty of Management and Economics, Tomas Bata University in Zlín, Mostní 5139, 760 00 Zlín, Czech Republic; 2Faculty of Management, University of Prešov in Prešov, Konštantínova 16, 080 01 Prešov, Slovakia; viera.ivankova@smail.unipo.sk (V.I.); martin.rigelsky@smail.unipo.sk (M.R.); ladislav.suhanyi@unipo.sk (L.S.)

**Keywords:** competitiveness, global competitiveness index, health inequalities, causes of mortality, gender, OECD

## Abstract

The aim of the present study is to quantify the relations between gender inequalities in mortality by selected causes of mortality and between competitiveness of Organisation for Economic Co-operation and Development (OECD) countries. Data for the analyses were obtained from OECD databases and the World Economic Forum (Global Competitiveness Index), for the years 2011–2016, for all 36 countries. The methods of descriptive analysis, analysis of differences in causes of mortality by gender characteristics, regression analysis, relationship analysis of gender inequalities in causes of mortality and competitiveness, and cluster analysis were used for the statistical data processing. Based on the research findings, it can be concluded that gender inequality occurs in most of the examined mortality groups of diagnoses, while it was most significant in the case of mortality due to neoplasms. The impacts of mortality on competitiveness are significant. In assessing gender inequalities in causes of mortality, significant impacts were seen in most mortality causes, but the most significant impact was identified within mortality due to neoplasms. Emphasis should be placed on men when reducing inequalities. Health and health equity should be supported by national governments, and health equity should be one of the key performance indicators of the country.

## 1. Introduction

Good health is a major source of economic and social development, therefore improving population health and reducing health inequalities represent one of the strategic objectives of the Health 2020 [1]. Substantial inequality in health is visible within the population divided into different groups according to income, education, gender or migrant status [2]. Health inequalities affect the society as a whole and we support the idea that these are unfair inequalities that can be avoided under certain conditions [3], even though long-term efforts are needed for their elimination [4]. Given the close relationship between public health and the processes of economic, social, and technological change [5], some opinions claim that reducing health inequalities can be beneficial for achieving development goals and for the overall economic progress of countries [6,7,8,9]. This implies that health inequalities should be among the main criteria in discussions on the sustainable development of countries, and policy makers should focus on reducing health inequalities with the purpose of reaching economic benefits [5]. The present study examines the issue in terms of the competitiveness of countries, as an important economic indicator of the economic condition of countries. The main purpose of the study is to evaluate the relations between gender inequalities in health and between competitiveness in developed Organisation for Economic Co-operation and Development (OECD) countries. The health issue is specified by mortality as a commonly used health indicator, which reflects the health conditions of a population and is a representative indicator of a country’s development.

## 2. Theoretical Background

There are many studies that deal with the mortality rate as a key indicator to measure inequalities in the health of a population [10,11,12,13,14]. Based on these findings, a decrease in mortality can be concluded, however the development of inequalities in mortality over time is not clear enough [15,16]. The perception of mortality inequalities is sensitive to the choice of method of differentiation, namely whether they are evaluated in absolute or relative terms [15,17]. In general, the development trend of mortality inequalities can be assessed as positive [18]. The development of mortality inequalities is influenced by many factors that have interactions in several areas of the social system, and therefore an examination of this issue in the economic dimension is essential.

Inequality in mortality is discussed in many studies from different perspectives; from a socio-economic point of view, it can be stated that the part of the population with lower incomes and lower education levels has a higher mortality rate [19], confirming the fact that socio-economic inequalities are an important factor of premature deaths [20]. Recent findings revealed that income levels in countries are an important factor in health inequalities [21]. On the other hand, it was pointed out that despite the strong relationship between income and health, there is no link between the changes in income inequalities and the changes in health inequalities [11]. At this point, it is also necessary to build on the fact that the improvement of population health may not depend on a better distribution of income in reducing income inequality [22]. The level of education is also included among the other socio-economic determinants of the health of the population creating health inequalities [23]. Many research findings declare that with an increase in education levels, mortality decreases, but the absolute health inequality increases as well [24]. Health inequalities have been identified between different population groups (e.g., ethnic groups [25]), between population groups differentiated by geographical area [26], between age groups or between occupation-based social classes [27]. 

With a focus on gender inequalities in mortality, many studies have dealt with and examined this issue [17,28]. Many research findings indicate that men achieve higher mortality rates than women [29,30]. Over time, it is possible to talk about a decrease in the mortality of both women and men; despite this fact, the relative gap between female and male mortality has widened and absolute inequalities narrowed [16]. In assessing the reduction of inequalities in mortality for both genders, Marshall-Catlin et al. [31] confirmed that the decrease in mortality over time was higher in the case of men than women, regardless of income or education level. 

Gender inequalities can be seen not only in all-cause mortality, but also in specific causes of death; for example, represented by respiratory mortality, in which men achieve higher mortality rates [32], resulting in better survival chances for women in obstructive airways disease [33]. A strong unfavorable prevalence of men over women is also evident in cancer mortality, where the mortality of women has been shown to be lower in 13 of the 18 cancer types [34]. Similar negative results for men were also found in the case of colorectal cancer [35]. On the other hand, it is necessary to point out the reduction of gender inequalities in chronic obstructive pulmonary disease (COPD) and lung cancer mortality, which means that the mortality of females due to these causes is close to that of men, although men are still at a disadvantage in this gender comparison [36,37]. Kriksic et al. [38] discovered that the rate of cardiovascular mortality is higher in women than in men, while this increased mortality was mainly associated with obesity, physical inactivity, and socio-economic factors. Following this fact, it should be noted that women with diabetes have a higher risk of cardiovascular disease as well as all-cause mortality compared to men [39,40]. Looking at stroke, it is not possible to talk about clear evidence, since research findings revealed that women under the age of 75 have a lower risk of stroke than men [41]. On the other hand, women of higher age overtake men more and more with increasing age, during the period of life when the risk of stroke in humans is the highest [42]. Similar results were shown in the study of Ayala et al. [43], who confirmed that age-specific rates of stroke mortality were lower in the case of women aged between 25–64 than in the case of men, but women aged 65 and over had higher mortality rates for ischemic stroke. Other research findings point out that the rate of stroke mortality is higher to the detriment of men [44]. 

Many studies consistently point out the fact that gender inequalities in health are persistent, they are not explained sufficiently, and they are formed by multi-level social factors [45]. This issue has also been studied in detail in the area of gender inequalities in mortality, as can be seen in the review of research studies mentioned above, but their interconnection with the economic dimension within the competitiveness of countries is still a poorly researched area. Several studies examined the relationship between mortality and economic benefits, while Boisclair et al. [46] confirmed that reducing mortality due to cardiovascular disease brings potential savings in terms of years of life saved. Looking at this issue from the other side, Kozlova et al. [47] assessed economic losses caused by population mortality. Reducing mortality has a positive impact on many aspects of economic life in countries, such as schooling and consumption [48], development [49], innovation [50], and transmission of knowledge [51]. One of the following important economic indicators—through which the relation with mortality can be examined—is a country’s competitiveness. In recent years, several studies have addressed the competitiveness of countries [52,53,54,55]. Many studies use the Global Competitiveness Index (GCI) index [56,57,58] when analyzing the competitiveness of countries, which is defined by the World Economic Forum (WEF) [59] as an annual yardstick for policy makers to look beyond short-term and reactionary measures and to instead assess their progress against the full set of factors that determine productivity. These are organized into 12 pillars: institutions, infrastructure, information and communications technology (ICT) adoption, macroeconomic stability, health, skills, product market, labor market, financial system, market size, business dynamism, and innovation capability. It is important to add that the GCI reporting methodology changed in 2018. Instead of a rating scale in the range of 1–7, used up until 2017 (inclusive), a new scale in the range of 1–100 was implemented the following year. The economic link between human immunodeficiency virus/acquired immunodeficiency syndrome (HIV/AIDS) mortality and human capacity in the context of global competitiveness has been indicated and confirmed in the literature [60]. On the other hand, Bucher [61] revealed a high correlation between a country’s rating based on GCI index and the level of gender inequalities, as well as the human development index (HDI), in which one of its constructs is the country’s health indicator (life expectancy). 

The above-mentioned facts motivated us to carry out research, the main objective of which was to quantify the relationships between gender inequalities in mortality by selected causes of mortality and the competitiveness of OECD countries. 

## 3. Materials and Methods 

The aim of the present study is to quantify the relations between gender inequalities in mortality by selected causes of mortality and between competitiveness of OECD countries. We chose several analytical trajectories to achieve this aim. In the first phase, inequality in selected causes of mortality between men and women is assessed, followed by the quantification of the impact of mortality of women and men on the competitiveness of OECD countries as well as the differences between women and men. Since we were also interested in the position of individual OECD countries, in the next phase we focus on assessing the position of OECD countries using cluster analysis.

For the analytical processes we used data from the OECD database [62] and from Global Competitiveness Reports published by the World Economic Forum (from 2011–2012 to 2016–2017). The indicators of the causes of mortality (CM) consisted of 13 variables (each in variation for men and women) and we focused on: certain infectious and parasitic diseases (INFC), neoplasms (CNCR), diseases of the blood and blood forming organs (BLOD), endocrine nutritional and metabolic diseases (MTBL), mental and behavioral disorders (MNTL), diseases of the nervous system (NRVS), diseases of the circulatory system (CRCL), diseases of the respiratory system (RSPT), diseases of the digestive system (DGST), diseases of the skin and subcutaneous tissue (SKIN), diseases of the musculoskeletal system and connective tissue (MSCL), diseases of the genitourinary system (GNTR), and certain conditions originating in the perinatal period (PRNT). All variables were reported in deaths per 100,000 population at standardized rates. These variables can be defined as age-standardized death rates per 100,000 population for selected causes calculated by the OECD Secretariat, using the total OECD population for 2010 as the reference population [63]. 

The competitiveness of economies was examined through the Global Competitiveness Index (GCI). This indicator takes values in the range of 1–7, where a higher number represents a more positive value of the competitiveness of economies. All OECD countries with current full membership were included in the analyses: Australia (AUS), Austria (AUT), Belgium (BEL), Canada (CAN), Czech Republic (CZE), Denmark (DNK), Estonia (EST), Finland (FIN), France (FRA), Germany (DEU), Greece (GRC), Hungary (HUN), Chile (CHL), Iceland (ISL), Ireland (IRL), Israel (ISR), Italy (ITA), Japan (JPN), Korea (KOR), Latvia (LVA), Lithuania (LTU), Luxembourg (LUX), Mexico (MEX), Netherlands (NDL), New Zealand (NZL), Norway (NOR), Poland (POL), Portugal (POR), Slovak Republic (SVK), Slovenia (SVN), Spain (ESP), Sweden (SWE), Switzerland (CHE), Turkey (TUR), United Kingdom (GBR), and the United States (USA). Due to the high frequency of missing data since 2017, data from 2011 to 2016 were used. 

In the first step, a descriptive analysis was applied along with an analysis of differences, that included commonly used statistical indicators and a non-parametric difference test of two independent samples using the Wilcoxon rank sum test (W). This part of the analysis was applied in the data of the causes of mortality of men and women. The correct test was selected based on the normality calculated using the Shapiro–Wilk test (S-W). These actions were applied in order to emphasize and point out the existence of differences in health between men and women, as well as in order to become more familiar with the variables included in the analyses. 

It was followed by the use of regression analysis, in which the variables of causes of mortality in men and women as well as inequality between men and women in absolute values were used as independent variables. Before the application of multiple linear regression, prediction tests were applied to determine the most appropriate model. The significance of the time series effect was tested by the F-test for individual and/or time effects. The presence of outliers was also tested using the Bonferroni outlier test [64]. Based on the Gauss–Markov theorem, it can be stated that the absence of significant heteroscedasticity and the lack of multicollinearity are dominant assumptions for the application of the Best Linear Unbiased Estimator (BLUE) in a large sample. Thus, heteroscedasticity and multicollinearity are considered the most important assumptions of the regression model. Multicollinearity was assessed using the variance inflation factor (VIF) method. The homogeneity of residue variability (homoscedasticity) was verified by the Breusch–Pagan test. If the assumptions are met, a multiple linear regression model is used. If significant heteroscedasticity is found, the coefficients will be estimated using the HC3 estimator [65]. The application of regression analysis and all the above-mentioned procedures represent the most important part of analytical processing in terms of the aim of the study. This part of the analytical processing conveys information about the existence, non-existence, and intensity of the effect between a particular health indicator and competitiveness. 

The final part of the analysis was devoted to cluster analysis (agglomerative hierarchical clustering). In the first step, the data entering this analysis were adjusted by the arithmetic mean for all years. Next, the data were standardized from 0 to 1, where 1 means a more positive rating. In the subsequent analytical steps two new variables were created: assessment of the gender inequalities in cause of mortality (GENDER_INEQ) and assessment of Global Competitiveness Index (EVAL_GCI). Silhouette (for average silhouette width) method was used to estimate the optimal number of clusters. Ward’s method was used to estimate the clusters themselves; this method was selected based on the highest agglomerative coefficient. At the end, a correlation analysis was applied using the Spearman coefficient ρ, which was selected based on the Royston’s multivariate normality test. The highest added value of the analysis described above lies mainly in conveying information from the applied point of view of research. The outputs specifically point to countries where the relationship between the evaluation of health indicators and the evaluation of competitiveness is perceived with positive intentions and, conversely, they will also point to countries where there are significant possibilities for improvement. 

The programming language R v. 3.6.1 (RStudio, Inc., Boston, MA, USA) (nickname: Action of the Toes) was used for analysis in the R Studio environment.

## 4. Results

### 4.1. Descriptive Analysis 

In order to increase the added value for readers, the appendix includes Table A1, which provides the average values of all analysed variables in each country. Based on the outputs in this table, the most positive values of GCI can be identified in Germany, Finland, Netherlands and the United States (mean = 5.5) and the most negative value was found in Greece (mean = 4.0). It can also be identified that Mexico showed the most negative values of several health indicators (mean: BLOD = 15.1; MTBL = 360.9; DGST = 188.4; MSCL = 15.3; PRNT = 15.2), on the other hand, Mexico acquired the most positive value of the CNCR variable (mean = 258.2). From the opposite point of view, Japan showed the most positive values of several health indicators compared to other countries (mean: NRVS = 26.1; CRCL = 319.8; PRNT = 1.4). Focusing on Finland, the country showed the most positive values in the INFC variable (mean = 9.7) and in the RSPT variable (mean = 64.1), but the most negative value was found in the NRVS variable (mean = 213.5). These outputs may provide an additional overview of selected indicators, but at this point it is appropriate to focus on the analytical process of this research.

In the first step of the following analytical procedure, we focused on the test of difference (Dif test), which is shown in the last part of Table 1. When applying and selecting a correct test, it is essential to meet the assumptions where the primary focus is on normality. Based on the results of the Shapiro–Wilk test, we cannot speak of the fulfilment of the normality assumption. 

In the test itself, the *p*-value is the most significant, where a significant difference was confirmed in most cases (the difference was not confirmed in only two cases—MNTL and SKIN). In this step, we focused on the individual causes of mortality. For easier illustration of the outputs we used average values. The INFC variable tells us that about 5.5 less women than men died of certain infectious and parasitic diseases, calculated per 100,000 inhabitants on average. The most significant difference was seen in the CNCR variable, in which approximately 112.7 less women than men died of neoplasms disease, calculated per 100,000 inhabitants. Similarly, we also compared other diagnoses showing that in all but one (MSCL) causes, there was a significant difference in mortality for the benefit of women. It is logical that more men die in diseases of the musculoskeletal system and connective tissue, as this disease is the result of hard physical activity more typical of men in most cases. Another very valuable indicator of the statistical characteristics of the monitored sample is the standard deviation (SD), which makes sense mainly in comparison between men and women. In most cases, a lower value was measured for women, which means that the yearly mortality rates in individual countries differ less for women than men. In relation to men, a lower value was measured only in two cases: diseases of the nervous system (NRVS) and diseases of the musculoskeletal system and connective tissue (MSCL). A 95% confidence interval was also calculated to give a more specific idea of the characteristics. The analyses also included a variable determining the competitiveness of economies, the Global Competitiveness Index (GCI), which acquired an average value of 4.893 for the monitored period and a median value of 4.985. As can be seen, the selected values of descriptive statistics differ slightly, so there is a presumption of outliers. The standard deviation is about 0.4970, the skewness is about −0.301, and the kurtosis is about −0.713, which do not indicate significant deviations from the normal statistical distribution.

### 4.2. Analysis of Effects—Regression Analysis and Correlation Analysis 

In the next step, regression analysis was used, which exactly confirmed and defined the existence of statistically significant independent variables affecting competitiveness. Testing took place in three areas: female mortality (model female), male mortality (model male), and gender inequalities in mortality (model gender inequalities). Before applying the regression analysis, it is advisable to assess the assumptions. Multicollinearity of independent variables is a very important assumption. The VIF method has been used for its assessment, where none of the independent variables had a value greater than 10 and most variables had a VIF value less than 5 (a value between 5 and 10 was acquired by four variables). We assessed the observed multicollinearity rate as acceptable. In addition to this test, other tests have been used: tests for the evaluation of the effect of time—F-test for individual and/or time effects (which indicates that the effect of time in years is not significant); test of outliers—Bonferroni outlier test (which identifies several outliers that have been removed—Switzerland 2012 and 2013); residue variability constancy test—Breusch–Pagan test (which reveals the presence of significant heteroscedasticity). The estimator HC3 is used to estimate the coefficients.

All three models in Table 2 can be evaluated as significant. With a focus on the model female, it has a multiple R^2^ value of approximately 0.6657 and an adjusted R^2^ value of 0.6425, which can be evaluated as sufficient. If we look at the coefficients themselves, it is possible to say that the competitiveness is significantly influenced by three variables determining the causes of mortality of women. These are the mental and behavioral disorders (MNTL β = 0.0189, *p* < 0.001), where the area of female mortality is very significant and at the same time its coefficient is positive; therefore, higher mortality due to this disease has a positive effect on the competitiveness of economies. Diseases of the circulatory system (CRCL β = −0.0018, *p* < 0.001) and diseases of the respiratory system (RSPT β = −0.0071, *p* < 0.001) have negative coefficient values, therefore lower mortality due to this disease has a positive effect on competitiveness.

Looking at the model male, with a multiple R^2^ value of 0.6957 and an adjusted R^2^ value of 0.6793, which can be evaluated as sufficient, it is obvious that men show a significant impact in more areas of causes of mortality than in women. The most significant impact was noticed in mental and behavioral disorders (MNTL β = −0.02, *p* < 0.001), where it takes a positive coefficient value, which can be understood as increasing the competitiveness of economies if men’s mortality increases. It is also possible to see a positive effect in certain infectious and parasitic diseases (INFC β = 0.0101, *p* = 0.008). Significant effects on competitiveness were also reflected in the variables neoplasms (CNCR β = −0.0048, *p* < 0.001), certain conditions originating in the perinatal period (PRNT β = −0.0844, *p* < 0.001), diseases of the blood and blood forming organs (BLOD β = −0.1078, *p* = 0.004), and diseases of the skin and subcutaneous tissue (SKIN β = −0.0667, *p* = 0.002); all with the output of negative coefficients, thus reducing the mortality of men in the above-mentioned areas will have a positive effect on the competitiveness of economies. The last part of the table shows the impact of gender-based inequalities by selected areas of mortality on the competitiveness of economies. Multiple R^2^ is 0.6427 and adjusted R^2^ is 0.6174. This is a sufficient measure of the determination coefficient. As can be seen, there was a significant difference in most areas. The most significant effect was seen in the neoplasms (CNCR β = −0.0095 ^†^) with a negative coefficient output, so if the inequality in the given cause of mortality decreases the competitiveness is expected to increase. 

Significant areas of gender CM inequalities with a negative coefficient include diseases of the digestive system (DGST β = −0.0145, *p* < 0.001), diseases of the skin and subcutaneous tissue (SKIN β = −0.1765, p = 0.018), diseases of the genitourinary system (GNTR β = −0.0314, *p* < 0.001), and certain conditions originating in the perinatal period (PRNT β = −0.2920, *p* < 0.001); here it applies that if the gender inequalities in the number of deaths decrease, then the competitiveness increases. A negative coefficient also appeared within diseases of the musculoskeletal system and connective tissue (MSCL β = −0.0676, p = 0.065), although the impact of this area is significant only at the level of α = 0.1. Significant impacts with a positive coefficient can be observed in certain infectious and parasitic diseases (INFC β = 0.0316, *p* < 0.001), mental and behavioral disorders (MNTL β = 0.0381, *p* < 0.001), diseases of the nervous system (NRVS β = 0.0385, *p* = 0.002), and diseases of the respiratory system (RSPT β = 0.0079, *p* = 0.001); thus, when inequality increases, then competitiveness also increases.

Table 3 offers a bivariate view of the competitiveness of economies and selected areas of mortality for women, men, and gender inequalities.

The most significant link between female mortality and competitiveness can be seen in the mental and behavioral disorders (MNTL ρ = 0.67, *p* < 0.001). This is a positive value, indicating that the increase in mortality from the disease affects the growth of competitiveness. The most significant link between male mortality and competitiveness can be seen in mental and behavioral disorders (MNTL ρ = 0. 64, *p* < 0.001). Similar to women it is a positive value, which can be interpreted analogously to women. When focusing on gender inequalities, neoplasms (CNCR) showed the highest relation (ρ = −0.51, *p* < 0.001). All significant coefficients in this area are negative, therefore the competitiveness increases by the decrease of inequality in mortality of men and women. This output will also serve as a basis for subsequent cluster analysis assessing the individual countries in connection to gender mortality inequalities and competitiveness.

### 4.3. Cluster Analysis

The following part is devoted to the application of cluster analysis (agglomerative hierarchical clustering), which aims to determine the countries based on the assessment of the outcomes of gender mortality inequalities and competitiveness. The analyses were entered by data, which were adjusted by the arithmetic mean for all years in the first step. In the next step, the data were standardized from 0 to 1, where 1 means a more positive evaluation. In the last step of adjustment, the data were adjusted by the arithmetic mean of each latent variable. In terms of gender mortality inequalities, this analysis included causes of mortality showing a causal relationship with competitiveness (Table 2) as well as a bivariate relationship (Table 3). These causes of mortality include: INFC, CNCR, MNTL, NRVS, CRCL, RSPT, DGST, SKIN, GNTR, and PRNT. In order to carry out the above-mentioned analyses, we created two new variables: the average value of selected areas of assessment of inequalities in individual countries for the reporting period (GENDER_INEQ) and the average value of the countries’ competitiveness for the reporting period (EVAL_GCI). The silhouette method (for average silhouette width) was used to estimate the optimal number of clusters. This method recommends two clusters for the relation of gender mortality inequalities and competitiveness. Based on the output of the agglomerative coefficient (0.9690) the Ward’s method appears to be most appropriate. The first cluster consists of the following countries: Australia, Austria, Belgium, Canada, Denmark, Germany, Finland, France, United Kingdom, Switzerland, Ireland, Israel, Iceland, Japan, Korea, Luxembourg, Netherland, Norway, New Zealand, Sweden, and the United States. The second cluster consists of Czech Republic, Spain, Estonia, Greece, Hungary, Chile, Italy, Latvia, Lithuania, Mexico, Poland, Portugal, Slovak Republic, Slovenia, and Turkey. The first cluster shows an average value of mortality assessment (GENDER_INEQ) of 0.7123 and an average value of Global Competitiveness Index (EVAL_GCI) of 0.8081. From the perspective of the analyzed variables, we evaluate the countries in this cluster very positively. The second cluster shows an average value of mortality assessment (GENDER_INEQ) of 0.5656 and an average in competitiveness (EVAL_GCI) of 0.2857. It is evident that the countries in the second cluster show lower rates of assessment of the analyzed variables, compared to the first cluster.

Figure 1 shows the interconnection between GENDER_INEQ and EVAL_GCI in color resolution of clusters. Countries that have achieved positive evaluation of competitiveness are also achieving positive values when evaluating mortality. For completeness and completion of the idea, a relationship analysis was applied to the variables. The choice of the most appropriate method was determined by the multivariate normality, which was verified by Royston’s multivariate normality test. This test, based on the H value (10.1319), defines a *p*-value approximately equal to 0.006, so it is appropriate to consider that the deviations from the normal distribution are significant. Therefore, the non-parametric Spearman’s ρ method was used to assess the relationship, which at the *p*-value of 0.0026 is calculated at 0.4912, what can be interpreted as a medium or substantial degree of correlation.

## 5. Discussion

Health systems should aim to improve the health of the population while making efficient use of the sources of funding to ensure the sustainability of the system. Healthcare is a sector in which the allocation of increased financial resources does not automatically improve the health of the population [66]. Therefore, many tools are used to assess the health policy effectiveness and the effectiveness of interventions implemented in health systems; these tools support health research through defined health metrics in the context of demographic parameters. In addition to their importance in quantifying regional health disparities, they also support the development of international benchmarks. This is the reason why the present research is focused on the examination of selected determinants of population health in terms of the economic condition of developed countries that allows us to create multi-dimensional evaluation research lines. Mortality represents a frequently used health indicator to define the health of a population [11,12,13,14] and this study deals with 13 diagnosis groups related to the causes of mortality. 

### 5.1. Descriptive and Difference Analysis Output—Gender Health Inequalities

As can be seen in the outputs of difference analysis, the highest rates of mortality in both genders were achieved by the diagnosis groups of neoplasms (CNCR) and diseases of the circulatory system (CRCL) [67,68]. The research revealed no significant gender difference in the diagnosis groups of mental and behavioral disorders (MNTL) and diseases of the skin and subcutaneous tissue (SKIN). This can be considered positively in terms of health equity. The difference between the other diagnosis groups has been confirmed, revealing the gender inequality in health as a negative reflection of health and social policies [17,28]. In the case of quantified significant difference, more positive results were found mainly in the female population. These results are consistent with many other findings [29,30,32,34,35,36,37,69]. In the male population, more positive values were found only in the case of diseases of the musculoskeletal system and connective tissue (MSCL). 

### 5.2. Regression and Correlation Analysis Output—Relation between Health and Economic Output

Focusing on the outputs of a regression analysis, a significant effect of gender inequalities was noticed in almost all examined diagnosis groups (INFC, CNCR, MNTL, NRVS, RSPT, DGST, SKIN, MSCL, GNTR, PRNT). The coefficients acquire negative values in the vast majority of cases, on the basis of which it is possible to recommend a reduction in these inequalities with a view to increase the competitiveness of economies. Positive values of the coefficient occurred within significant relations in the case of INFC, MNTL, NRVS, and RSPT. This output can be interpreted as meaning that if these inequalities deepen, they will have a positive effect on competitiveness. This can be explained by (i) the nature of the diseases, the cost of their treatment, the quality of life after treatment and others, or by (ii) the application of a multivariate approach itself, so certain variables interact with each other, causing the given outcome.

When interpreting the impacts on competitiveness, it is appropriate to focus primarily on models of men and women. Men acquire significant effects in the following diagnosis groups: INFC, CNCR, BLOD, MNTL, SKIN, GNTR, and PRNT. In the case of women, there were fewer groups with significant effects, namely, MNTL, CRCL, and RSPT. It can be concluded that the efforts to reduce the incidence of disease in order to increase competitiveness will be more effective on the part of men than on the part of women.

With a focus on specific diagnosis groups that acquire significantly negative rates of gender inequality and a significant rate of the effect in at least one variant of the gender characteristic model, the following can be stated: the increase in competitiveness will be expected when the number of diseases and the frequency of mortality per CNCR, SKIN and PRNT diagnosis groups decrease in the case of men. The greatest efforts should be devoted to diseases of the CNCR diagnosis group.

The bivariate view of the relations between the outputs of health indicators and competitiveness determined by correlation analysis acquires similar values as the previous regression analysis in several indicators. A significant relationship was observed in the diagnosis groups CNCR, CRCL RSPT, DGST, and PRNT from the perspective of gender inequality in health. The coefficient was negative in all cases. Consequently, reducing inequalities assumes an increase in competitiveness. From the point of view of reducing inequalities, it makes sense to focus on specific primary diagnosis groups of men and women, where a significant relationship has been revealed in gender inequalities. From the point of view of increasing competitiveness, in the case of CNCR it makes more sense to focus on reducing inequalities particularly in men. The highest degree of correlation was measured in the case of CNCR, which can be interpreted as moderate to significant. From the point of view of the interconnection of competitiveness, the model male acquires higher and statistically more significant coefficients for RSPT, DGST and PRNT. On the contrary, CRCL predominates in women. 

These findings confirm the importance of reducing health inequalities in order to achieve economic benefits [70], as health inequalities are considered a huge economic burden [7].

### 5.3. Cluster Analysis Output—Relation between Health and Economic Output

As the results of cluster analysis show, countries such as the United States, Finland, and Sweden are assessed as the countries with the most positive outputs in terms of the evaluation of the analyzed health and competitive indicators. On the other hand, countries such as Latvia, Lithuania, Estonia, and Mexico showed least positive outputs, indicating opportunities for improvements. These countries should be focused on improving their health outcomes as well as competitiveness. In general, countries with lower income rates, respectively with a lower proportion of higher-income groups, showed more negative outcomes. 

This creates opportunities for us to investigate economic parameters, such as income and its impact on health. Gender inequalities are created by society and therefore can be changed. The status of the women has improved over the past century, but not in all countries; there are many new challenges for the improvement of life and the elimination of gender inequalities. Health inequality can also be related to income inequality [21], which determines the socio-economic position of individual men and women. Research shows that the distribution of health and disease in the population follows a social gradient: the lower the socio-economic position, the worse the health status [19,20,21,23,24]. 

## 6. Conclusions

As pointed out, the existence of health inequalities is an undeniable fact that also occurs in developed countries. Public health is a very complex apparatus, the fluctuations of which are reflected in the economies of countries and affect them. Health of society is an important indicator in the development of a country, and both poor and rich countries have to face many health problems. Their solution differs depending on the health care systems, the degree of health protection provided to disadvantaged people, and the level of health care funding. Social stratification determines a differentiated approach to health care, as well as the use of health care services, which requires building health care systems based on the principles of equity, disease prevention, and health promotion. The competitiveness of a country is closely related to these factors. The aim of the present study was to assess the relations between gender inequalities in mortality and between competitiveness in developed OECD countries. 

This aim was achieved through several procedures. Using the characteristics of descriptive statistics and analysis of differences, the existence of gender inequalities was pointed out, within which women gained more positive outcomes. The regression analysis revealed in several cases the significance of the effects of specific health indicators on competitiveness in the specification of men, women, as well as gender inequalities. The correlation analysis also showed a relation between the mentioned areas. The application of cluster analysis classified the countries in terms of evaluating economic and health output, where the positively evaluated countries were identified, as well as the countries with lower evaluation, where there is possibility of improvement. 

As mentioned, the higher mortality rate from the majority of the examined diagnoses groups was in men. If policies focus on increasing prevention measures primarily in men, a reduction in gender health inequalities and an increase in countries’ competitiveness can be expected. At this point, it is possible to emphasize the importance of promoting a healthy lifestyle, as well as the elimination of smoking and alcohol consumption. A very high proportion of deaths is due to CNCR diseases, most of which are classified as preventable mortality. Preventive policies in this area would have a massive impact on increasing the competitiveness of economies.

These findings are very important for policy makers whose activities should aim to eliminate gender inequalities in health. It is also important to assess the impact of individual policies and programs on health and health equity, and thus to permanently examine the determinants that have an impact on regional health disparities and the level of their impact or regulatory potential. Every aspect of the country’s government and economy has a strong potential to affect health, or health inequality. Activities in the scope of social determinants of health also require building and strengthening capacities in health care. It is therefore essential to create basic databases to monitor health inequalities, which could contribute to the development of effective policies, systems, and programs. It is important to develop health care systems based on the principles of equity, disease prevention, and health promotion.

Among the limitations that need to be mentioned, the research sample consisted of developed countries; therefore, the results are only generalizable to developed countries. In addition, a certain time series was included in the analysis; its selection was conditioned by maximizing credibility/representativeness and at the same time by minimizing the external effects of time. The chosen time period can be considered sufficient from the point of view of representativeness and at the same time it can be assumed that the hidden effects of time did not significantly affect the outputs.

Future research will focus on other applications of economic relations and gender-oriented inequalities, since this issue is only vaguely examined and therefore any scientific efforts are welcome. Specifically, we plan to target different groups of countries, including the less developed ones, as well as incorporate other health metrics. We also seek to investigate indicators that are suitably linked—from the point of view of processes and data—to mortality indicators, such as potential years of life lost (PYLL), quality adjusted life year (QALY), and others. This makes it possible to link them to the lost productivity and spending levels in countries’ health systems, which will create another new research dimension and the possibility of quantifying the effects of mortality on several spheres of socio-economic and social life.

## Figures and Tables

**Figure 1 ijerph-17-03698-f001:**
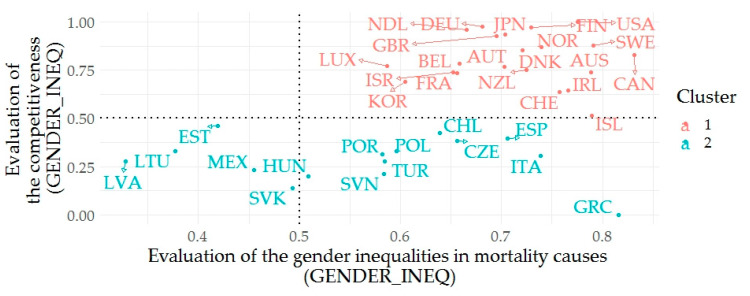
Relation between evaluation of gender inequalities in mortality causes and competitiveness.

**Table 1 ijerph-17-03698-t001:** Descriptive statistic and difference test.

		INFC	CNCR	BLOD	MTBL	MNTL	NRVS	CRCL	RSPT	DGST	SKIN	MSCL	GNTR	PRNT
F	Mean	11.4465	164.7535	2.3030	26.2900	24.5420	29.0495	249.7840	50.6580	26.9905	1.6475	4.5190	13.7245	2.3240
	SD	5.9057	26.4960	1.2777	28.0870	17.6477	17.6200	113.6978	20.4510	10.1736	1.3668	1.7802	6.4830	1.3221
	CI 95%	10.6230	161.0589	2.1248	22.3736	22.0812	26.5926	233.9302	47.8064	25.5719	1.4569	4.2708	12.8205	2.1397
		12.2700	168.4481	2.4812	30.2064	27.0028	31.5064	265.6378	53.5096	28.4091	1.8381	4.7672	14.6285	2.5083
M	Mean	16.8945	278.1220	2.6580	32.4805	25.6770	34.2785	360.1790	89.4960	44.9180	1.5710	3.4065	19.7090	2.8765
	SD	8.2881	52.8338	1.4246	30.0712	14.8262	17.3646	179.6917	27.0570	21.4373	1.4238	1.3129	8.6967	1.6601
	CI 95%	15.7388	270.7549	2.4594	28.2874	23.6097	31.8572	335.1231	85.7232	41.9288	1.3725	3.2234	18.4963	2.6450
		18.0502	285.4891	2.8566	36.6736	27.7443	36.6998	385.2349	93.2688	47.9072	1.7695	3.5896	20.9217	3.1080
ABS F-M	Mean	5.5147	112.6588	0.5078	6.4348	4.3676	5.4348	110.0044	38.5426	17.7779	0.3050	1.2250	5.9495	0.6251
	SD	3.7728	44.6773	0.4167	3.6089	3.1358	2.6032	70.1434	17.1292	12.5011	0.2853	0.9524	3.0609	0.4512
	CI 95%	15.7820	273.1206	2.4889	28.7128	23.2596	31.4754	341.5075	86.1079	42.5928	1.3872	3.2472	18.7783	2.6880
		17.8847	286.8187	2.8581	36.4633	27.0438	35.8519	388.5404	93.0434	48.1841	1.7479	3.5878	21.0132	3.1129
Dif test	S-W (F)	0.9329 ^†^	0.9433 ^†^	0.8735 ^†^	0.5198 ^†^	0.9749 **	0.7540 ^†^	0.7825 ^†^	0.9692 ^†^	0.8154 ^†^	0.6618 ^†^	0.9858 *	0.8630 ^†^	0.7902 ^†^
	S-W(M)	0.8791 ^†^	0.9851 *	0.8520 ^†^	0.4961 ^†^	0.9473 ^†^	0.7678 ^†^	0.8358 ^†^	0.9828 *	0.7858 ^†^	0.7383 ^†^	0.9721 ^†^	0.9116 ^†^	0.8033 ^†^
	W	32972.00	21953.50	37892.50	35900.00	40496.50	36034.00	31227.50	26270.00	27795.00	40052.50	34346.50	31846.00	35295.00
	*p*-value	<0.001	<0.001	0.001	<0.001	0.305	<0.001	<0.001	<0.001	<0.001	0.524	<0.001	<0.001	<0.001

Notes: * *p*-value < 0.05; ** *p*-value < 0.01; ^†^
*p*-value < 0.001. Abbreviations: F, female; M, male; ABS, absolute value; S-W, Shapiro–Wilk normality test; W, Wilcoxon rank sum test; Dif test, test of difference; INFC, certain infectious and parasitic diseases; CNCR, neoplasms; BLOD, diseases of the blood and blood forming organs; MTBL, endocrine nutritional and metabolic diseases; MNTL, mental and behavioral disorders; NRVS, diseases of the nervous system; CRCL, diseases of the circulatory system; RSPT, diseases of the respiratory system; DGST, diseases of the digestive system; SKIN, diseases of the skin and subcutaneous tissue; MSCL, diseases of the musculoskeletal system and connective tissue; GNTR, diseases of the genitourinary system; PRNT, certain conditions originating in the perinatal period.

**Table 2 ijerph-17-03698-t002:** Regression models output.

Ordinary Least Squares (OLS)	Model Female		Model Male		Model Gender Inequalities	
	HC3		HC3		HC3	
	F = 28.62 (*p* < 0.001)		F = 32.53 (*p* < 0.001)		F = 25.46 (*p* < 0.001)	
	R^2^ = 0.67; R^2^ adj. = 0.64		R^2^ = 0.70; R^2^ adj. = 0.68		R^2^ = 0.64; R^2^ adj. = 0.62	
	Estimate	Pr (>|t|)	Estimate	Pr (>|t|)	Estimate	Pr (>|t|)
Intercept	5.3685	<0.001	5.9336	<0.001	5.8748	<0.001
INFC	0.0064	0.430	0.0101	0.008	0.0316	<0.001
CNCR	−0.0005	0.725	−0.0048	<0.001	−0.0095	<0.001
BLOD	−0.0592	0.296	−0.1078	0.004	−0.0464	0.430
MTBL	0.0021	0.329	−0.0015	0.446	−0.0031	0.708
MNTL	0.0189	<0.001	0.0200	<0.001	0.0381	<0.001
NRVS	0.0011	0.341	−0.0010	0.524	0.0385	0.002
CRCL	−0.0018	<0.001	0.0000	0.935	0.0006	0.329
RSPT	−0.0071	<0.001	0.0002	0.884	0.0079	0.001
DGST	0.0038	0.212	0.0013	0.297	−0.0145	<0.001
SKIN	−0.0444	0.156	−0.0667	0.002	−0.1765	0.018
MSCL	−0.0322	0.153	0.0240	0.333	−0.0676	0.065
GNTR	0.0112	0.141	0.0101	0.090	−0.0314	<0.001
PRNT	−0.0346	0.156	−0.0844	<0.001	−0.2920	<0.001

**Table 3 ijerph-17-03698-t003:** Analysis of relationship.

Spearman ρ	Female		Male		Gender Inequalities	
	ρ	Pr(>|t|)	ρ	Pr(>|t|)	ρ	Pr(>|t|)
INFC	0.1630	0.020	0.0318	0.652	−0.0650	0.356
CNCR	0.0456	0.518	−0.4105	<0.001	−0.5074	<0.001
BLOD	0.0983	0.162	0.1253	0.074	0.0334	0.635
MTBL	−0.1849	0.008	−0.1222	0.082	0.1155	0.100
MNTL	0.6665	<0.001	0.6434	<0.001	0.0963	0.170
NRVS	0.4546	<0.001	0.4739	<0.001	−0.0557	0.429
CRCL	−0.5215	<0.001	−0.4623	<0.001	−0.1989	0.004
RSPT	0.0578	0.412	−0.1624	0.020	−0.4133	<0.001
DGST	−0.2348	<0.001	−0.3685	<0.001	−0.4700	<0.001
SKIN	0.0667	0.344	0.0371	0.601	−0.1063	0.133
MSCL	0.261	<0.001	0.2725	<0.001	0.0129	0.855
GNTR	−0.043	0.541	−0.0498	0.479	−0.1085	0.122
PRNT	−0.2095	0.003	−0.2548	<0.001	−0.2994	<0.001

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
