# Peer review of "Impact of Gender Inequalities in the Causes of Mortality on the Competitiveness of OECD Countries"

_ijerph, 2020, doi:10.3390/ijerph17103698_

Round 1

Reviewer 1 Report

The problem to be studied is well presented and motivated, and also relevant scientifically as well as specifically for the journal. It also relies on an extensive review of relevant literature. Data are well selected, and the analyses relevant. The discussion and conclusion is well worked out.

My major problem, however, relates to the structure of the statistical analyses. Specifically, it is only stated in the introduction and methods section that "this and that statistical method will be used", but it is only when one comes to the results section it becomes clear why this and that method applies well. Rather, you should write in the introduction and methods section "in order to provide this and that analysis, we use this and that method", then it becomes clear from the beginning why the method is used, and the story will be more consistent.

Next, I also have problems with your English style. Not to say that you make explicit grammatic or other errata, but your sentence construction is not very fluent. I recommend having a good proofread, preferably by some native English speaking person, and ideally one from your scientific discipline. As it stands now, the language style is not acceptable for an international journal.

Some specific comments:

  • Be careful with your use of acronyms like CL_1 etc, Mean_GICM etc. It makes the presentation difficult to follow.
  • Make the tables easier to read. Replace acronyms like CLM_1 etc with short self-evident labels.
  • Any figures referred in the text should appear in a table. For example, R-Square values for the regressions; include a row in the table with them.
  • Clean up in your results and omit results that may be interesting for some or other reason, and include only those needed for the core presentation. For example, the figure with the dendrogram is not interesting for the presentation and should be omitted. Another example is the VIF factors reported in the text - omit them and just state that they were below the critical level and thus satisfactory. Etc...
  • In figure 2, you should write full length labels on axes, not acronyms, so that the figure is self-explaining.

Thus, to summarize, it is my guess that the paper is written by a promising, but unexperienced, scholar, who is less familiar with publication. If some of the authors are senior and more experienced, I strongly encourage them to take the responsibility for lifting the paper to scientific standard. Indeed, this is one of the most important responsibilities for those of us who are seniors.

I hope that a revision will be invited, and I wish you good luck with the work.

Author Response

Dear Reviewer,

First of all, we would like to thank all three reviewers for the time spent evaluating the manuscript. The comments were encouraging and very helpful in improving the study. We hope that the revisions improve the paper such that you now deem it worthy of publication in „International Journal of Environmental Research and Public Health“ and our revision has improved the paper to a level of your satisfaction. We have carefully revised the paper and rewrite accordingly. In the following we respond in detail to the all comments. All changes in the revised manuscript are highlighted in yellow.

Thank you for giving us the opportunity to revise our manuscript. We would like to send our revised study again for considering and we look forward to hearing from you.

Kind regards,

Beata Gavurova

Comments and Suggestions for Authors:

The problem to be studied is well presented and motivated, and also relevant scientifically as well as specifically for the journal. It also relies on an extensive review of relevant literature. Data are well selected, and the analyses relevant. The discussion and conclusion is well worked out.

Response - Thank you for the positive evaluation of our work.

My major problem, however, relates to the structure of the statistical analyses. Specifically, it is only stated in the introduction and methods section that "this and that statistical method will be used", but it is only when one comes to the results section it becomes clear why this and that method applies well. Rather, you should write in the introduction and methods section "in order to provide this and that analysis, we use this and that method", then it becomes clear from the beginning why the method is used, and the story will be more consistent.

Response - Thank you for the constructive criticism, all your recommendations have been accepted and included in the revised manuscript. The methods section has been supplemented with information that approximates the reason for the application of individual statistical processes.

Next, I also have problems with your English style. Not to say that you make explicit grammatic or other errata, but your sentence construction is not very fluent. I recommend having a good proofread, preferably by some native English speaking person, and ideally one from your scientific discipline. As it stands now, the language style is not acceptable for an international journal.

Response - Thank you for your comment and recommendation, based on which a language correction was made in terms of language style, but also grammar. The current version of the manuscript provides fluency in the text as a whole. English language has been improved.

Some specific comments:

Be careful with your use of acronyms like CL_1 etc, Mean_GICM etc. It makes the presentation difficult to follow.
Make the tables easier to read. Replace acronyms like CLM_1 etc with short self-evident labels.

Response - Thank you for your suggestions and recommendations. All abbreviations of health indicators have been replaced by more comprehensible abbreviations. We hope this change will make the study easier to read. 

Any figures referred in the text should appear in a table. For example, R-Square values for the regressions; include a row in the table with them.
Response - Thank you for your comment. We have incorporated it into the revised manuscript and thus R2 was added to the table.

Clean up in your results and omit results that may be interesting for some or other reason, and include only those needed for the core presentation. For example, the figure with the dendrogram is not interesting for the presentation and should be omitted. Another example is the VIF factors reported in the text - omit them and just state that they were below the critical level and thus satisfactory. Etc...

Response - Thank you for the constructive criticism, all your recommendations have been incorporated into the revised manuscript. Thus, the dendrogram was removed, specific VIF values were removed, and the results were interpreted more clearly and comprehensible.

In figure 2, you should write full length labels on axes, not acronyms, so that the figure is self-explaining.

Response - Thank you for the recommendation that helped to improve our manuscript. The figure has been modified as recommended.

Thus, to summarize, it is my guess that the paper is written by a promising, but unexperienced, scholar, who is less familiar with publication. If some of the authors are senior and more experienced, I strongly encourage them to take the responsibility for lifting the paper to scientific standard. Indeed, this is one of the most important responsibilities for those of us who are seniors.

I hope that a revision will be invited, and I wish you good luck with the work.

Response - Thank you for your review, your comments have been very useful for us and helped to improve the article, we appreciate them. We believe that we have met your expectations.

Reviewer 2 Report

This manuscript analyzes the impact of gender inequality mortality on the competitiveness of OECD countries. The manuscript is well-written and comprehensive in its analysis. At times the presentation becomes too elaborate and it is recommended the authors find ways to shorten certain sections.

Since several topics are presented in this manuscript, it would increase readibility to include corresponding subsections within the methods and results sections (and possibly discussion) so that a read can easily track individual topics such as the clustering work.

The mechanics of the methods (i.e. statistical, etc.) are well-described but more needs to be added to explain the reasoning for using the metrics, such as gender inequality. 

Section 1: This section and section 2 should be combined into an introduction. 

Section 2: This is a massive paragraph. I suggest describing CMs into groups so that the paragraph is easier to follow and read.

Lines 126-131: It is strange to have two 1-sentence paragraphs immediately following a paragraph that is almost the full length of two pages.

Line 155: Is this a typo? "Diseases of the Circulatory System 100000 F MSR"

Line 163: The GCI for each country should be listed in a table so the reader can view it. It would be useful to explain it in brief as well for readers not familiar with how it is estimated. It would also be good to list the mortality by CM on the same or a separate table.

Line 204: Methods was already listed as Section 4. This, and following sections, should be renumbered.

Lines 251 and 264: What does the "a" stand for in "Switzerland 2012 a 2013"?

Line 272: The multiple R-squared and adjusted R-squared value are not shown anywhere on Table 2. This (and the same issue in other tables) should be addressed.

Line 306: Is there a missing "=" between alpha and 0.1?

Line 330: In an earlier section, Table 3 was stated to be univariate but here it says bivariate. Which is it?

Line 342: "Such as" should be removed since the country list is presented.

Lines 441-444 seem to be repeated.

Line 452: Remove the word "like" as the country list is presented.

The Conclusion section restates many of the statements in the Discussion and should be more concise.

Author Response

Dear Reviewer,

First of all, we would like to thank all three reviewers for the time spent evaluating the manuscript. The comments were encouraging and very helpful in improving the study. We hope that the revisions improve the paper such that you now deem it worthy of publication in „International Journal of Environmental Research and Public Health“ and our revision has improved the paper to a level of your satisfaction. We have carefully revised the paper and rewrite accordingly. In the following we respond in detail to the all comments. All changes in the revised manuscript are highlighted in yellow.

Thank you for giving us the opportunity to revise our manuscript. We would like to send our revised study again for considering and we look forward to hearing from you.

Kind regards,

Beata Gavurova

Comments and Suggestions for Authors:

This manuscript analyzes the impact of gender inequality mortality on the competitiveness of OECD countries. The manuscript is well-written and comprehensive in its analysis. At times the presentation becomes too elaborate and it is recommended the authors find ways to shorten certain sections.

Response - Thank you for appreciating our work, and also for your comment that improved our manuscript. The recommendations were accepted, and several paragraphs were deleted from the analytical part:  

- Some interpretations of the descriptive statistics table

-  The process of testing the conditions of the use and selection of a regression model (it overwhelmed the reader)

-  Paragraphs from the conclusion, as we have evaluated the presented part as too long and several ideas can be derived from the previous parts.

We hope this will improve the readability of the study.

Since several topics are presented in this manuscript, it would increase readibility to include corresponding subsections within the methods and results sections (and possibly discussion) so that a read can easily track individual topics such as the clustering work.

Response - Thank you for the comment. The Results section was divided into three complex parts, which brings clarity and readability of the results to the study.

The mechanics of the methods (i.e. statistical, etc.) are well-described but more needs to be added to explain the reasoning for using the metrics, such as gender inequality. 

Response - Thank you for your comment, these methods are commonly used in researches in this field.

Section 1: This section and section 2 should be combined into an introduction. 

Response - Thank you for the recommendation, the individual sections are given by the template of the journal we have followed. The introduction section aims to introduce the reader to the main idea of the research. The theoretical background section offers specific findings of international studies and presents the current state of the problem based on their results.

Section 2: This is a massive paragraph. I suggest describing CMs into groups so that the paragraph is easier to follow and read.

Response - Thank you for the comment that helped to improve the quality of our study. The massive paragraph has been divided into several coherent parts, which improves the readability and clarity of scientific findings. Some specific diseases were grouped together.

Lines 126-131: It is strange to have two 1-sentence paragraphs immediately following a paragraph that is almost the full length of two pages.

Response - Thank you for the comment, by editing the manuscript we also decided to change the structure of the theoretical background section, which can provide the reader with a clearer overview of scientific findings in the field.

Line 155: Is this a typo? "Diseases of the Circulatory System 100000 F MSR"

Response - Thank you, this comment has been accepted and a correction has been made.

Line 163: The GCI for each country should be listed in a table so the reader can view it. It would be useful to explain it in brief as well for readers not familiar with how it is estimated. It would also be good to list the mortality by CM on the same or a separate table.

Response - Thank you for your recommendation, a table with country averages has been added to the attachments. The outline of the GCI index can be found in the theoretical background section, where its pillars are listed, as well as the scale for ranking. Please see lines 118-122.

Line 204: Methods was already listed as Section 4. This, and following sections, should be renumbered.

Lines 251 and 264: What does the "a" stand for in "Switzerland 2012 a 2013"?

Response - Thank you for the remark, it was fixed.

Line 272: The multiple R-squared and adjusted R-squared value are not shown anywhere on Table 2. This (and the same issue in other tables) should be addressed.

Response - Thank you for the comment, R2 was added to the tables, other selected values listed only in the text were deleted (e.g. VIF, etc.)

Line 306: Is there a missing "=" between alpha and 0.1?

Response - Thank you for the remark, the correction was made.

Line 330: In an earlier section, Table 3 was stated to be univariate but here it says bivariate. Which is it?

Response - Thank you for the remark, it was changed.

Line 342: "Such as" should be removed since the country list is presented.

Response - Thank you for the remark, it was removed.

Lines 441-444 seem to be repeated.

Response - Thank you for the comment, which helped to improve the quality of our study, it was incorporated into the manuscript edits. The discussion and conclusion sections have been considerably revised and simplified in order to improve the readability of the study as well as the main findings.

Line 452: Remove the word "like" as the country list is presented.

Response - Thank you for the remark, it was removed.

The Conclusion section restates many of the statements in the Discussion and should be more concise.

Response - Thank you for the recommendation, which has been incorporated into the revised manuscript. The conclusion section has been redesigned and now provides key findings without duplicating the discussion.

 - Thank you for your review, your comments have been very useful for us and helped to improve the article, we appreciate them. We believe that we have met your expectations.

Reviewer 3 Report

The Authors present a manuscript: "Impact of the gender inequalities in the causes of mortality on the competitiveness of OECD Countries." very interesting and original for its content  and type of study. The methodology followed strict and appropriate rules and the obtained results based on a large sample are very well reported in good tables.

I found the paper stimulating but I have to concerns:

  1. It is not easy to follow the final results that in part should be listed in a table that could summarize  and correlate them with the considered variables. Health inequality depends on: income, education, gender, migrant status. In the summary table creation or not correlation should be stressed in a clearer way. it is not easy for the reader to follow the text reporting variable to variable and their significant meaning.The same is for Mortality
  2. Discussion is too long and repeat the data reported in Results. It should be better to emphasize what the Authors found with their research and add something about the limitations of the research. In Conclusion I'd like to have a tempestive plan on how to overcome some of the raised problem especially about gender or do we have to passively  accept the results? Authors should make this effort for teaching something specifically to the readers.

Author Response

Dear Reviewer,

First of all, we would like to thank all three reviewers for the time spent evaluating the manuscript. The comments were encouraging and very helpful in improving the study. We hope that the revisions improve the paper such that you now deem it worthy of publication in „International Journal of Environmental Research and Public Health“ and our revision has improved the paper to a level of your satisfaction. We have carefully revised the paper and rewrite accordingly. In the following we respond in detail to the all comments. All changes in the revised manuscript are highlighted in yellow.

Thank you for giving us the opportunity to revise our manuscript. We would like to send our revised study again for considering and we look forward to hearing from you.

Kind regards,

Beata Gavurova

Comments and Suggestions for Authors:

The Authors present a manuscript: "Impact of the gender inequalities in the causes of mortality on the competitiveness of OECD Countries." very interesting and original for its content  and type of study. The methodology followed strict and appropriate rules and the obtained results based on a large sample are very well reported in good tables.

Response - Thank you very much for the positive evaluation of our work.

I found the paper stimulating but I have to concerns:

It is not easy to follow the final results that in part should be listed in a table that could summarize and correlate them with the considered variables. Health inequality depends on: income, education, gender, migrant status. In the summary table creation or not correlation should be stressed in a clearer way. it is not easy for the reader to follow the text reporting variable to variable and their significant meaning. The same is for Mortality

Response - Thank you for the comment, it helped to improve the quality of our study. The results section has been redesigned. We believe that the modifications will increase the readability of the study.

Discussion is too long and repeat the data reported in Results. It should be better to emphasize what the Authors found with their research and add something about the limitations of the research.

Response - Thank you for the recommendation, it has been accepted and incorporated into the revised manuscript. Both the discussion and the conclusions have been considerably redesigned and simplified in order to improve the readability of the study as well as the main findings. Limitations of the research has been added.

In Conclusion I'd like to have a tempestive plan on how to overcome some of the raised problem especially about gender or do we have to passively  accept the results? Authors should make this effort for teaching something specifically to the readers.

Response - Thank you for the recommendation that helped to improve our manuscript. One of the possible approaches to eliminate gender inequalities has been added. The importance of reducing inequalities in terms of countries' competitiveness was also emphasized.

Response - Thank you for your review, your comments have been very useful for us and helped to improve the article, we appreciate them. We believe that we have met your expectations.

Round 2

Reviewer 1 Report

The paper has improved substantially, and I feel that my requests have been met.

I have only a minor matter, merely one of convention: You report p-values with scientific notation in tables (for example 1.2 E-2). This is not very common. Rather report in digit values with, say, three digits (for example 0.214, 0.012, 0.002 etc.), and write "p<0.001" where p falls below this threshold. Also, in the text, I recommend reporting the statistical figure with the p-value (not the symbol indicating alpha level). For example, MNTL: beta=0.0189, p<0.001.

I am not insisting on the above; as a statistician, however, it is my preference. Therefore, I recommend "accept" and leave this and other details for communication with the editorial office, who is familiar with the standard of the journal.

Author Response

Dear Reviewer, 

Dear Reviewer, 

We appreciate your positive evaluation and we consider it to be a significant motivation for our further work. We have carefully considered your comments and incorporated them into the revised manuscript hoping that the changes are in line with your ideas. The responses to your comments and recommendations can be found in the following text and all changes in the revised manuscript are highlighted in yellow.

Both rounds of the reviewing process and your reviews have been inspiring, and we see that this has greatly improved the quality of our paper. We would like to thank you for the constructive feedback and for the opportunity to revise our manuscript. 

Kind regards,

Beata Gavurova

The paper has improved substantially, and I feel that my requests have been met.

Response - Thank you for the positive evaluation of our work.

I have only a minor matter, merely one of convention: You report p-values with scientific notation in tables (for example 1.2 E-2). This is not very common. Rather report in digit values with, say, three digits (for example 0.214, 0.012, 0.002 etc.), and write "p<0.001" where p falls below this threshold. Also, in the text, I recommend reporting the statistical figure with the p-value (not the symbol indicating alpha level). For example, MNTL: beta=0.0189, p<0.001.

Response - Thank you for your comment and recommendation, in all cases your suggestion has been included in the revised manuscript.

I am not insisting on the above; as a statistician, however, it is my preference. Therefore, I recommend "accept" and leave this and other details for communication with the editorial office, who is familiar with the standard of the journal.

Response - Thank you for your review, your comments have been very useful for us and helped to improve our article, we appreciate them.

Reviewer 2 Report

The authors incorporated feedback constructively into the manuscript. Two small suggestions:

1) The paragraph starting on Line 164 is nearly a page long. I suggest separating it into multiple paragraphs, perhaps grouping steps, but definitely not keeping it as is.

2) The appendix table was a necessary addition. I suggest explaining (in the caption) that the darker background means higher values for each column/health condition so it is more obvious to the reader.

Author Response

Dear Reviewer, 

We appreciate your positive evaluation and we consider it to be a significant motivation for our further work. We have carefully considered your comments and incorporated them into the revised manuscript hoping that the changes are in line with your ideas. The responses to your comments and recommendations can be found in the following text and all changes in the revised manuscript are highlighted in yellow.

Both rounds of the reviewing process and your reviews have been inspiring, and we see that this has greatly improved the quality of our paper. We would like to thank you for the constructive feedback and for the opportunity to revise our manuscript. 

Kind regards,

Beata Gavurova

The authors incorporated feedback constructively into the manuscript. Two small suggestions:

Response - Thank you for the positive evaluation of our work.

1) The paragraph starting on Line 164 is nearly a page long. I suggest separating it into multiple paragraphs, perhaps grouping steps, but definitely not keeping it as is.

Response - Thank you for your recommendation, the paragraph has been divided into several parts, which makes it easier to read.

2) The appendix table was a necessary addition. I suggest explaining (in the caption) that the darker background means higher values for each column/health condition so it is more obvious to the reader.

Response - Thank you for your recommendation, your suggestion has been included in the note to this table.

Response - Thank you for your review, your comments have been very useful for us and helped to improve our article, we appreciate them.